# Blood flow controls bone vascular function and osteogenesis

Saravana K. Ramasamy[1,2], Anjali P. Kusumbe[1,3], Maria Schiller[1], Dagmar Zeuschner[4], M. Gabriele Bixel[1], Carlo Milia[5], Jaba Gamrekelashvili[6], Anne Limbourg[7], Alexander Medvinsky[8], Massimo M. Santoro[5,9], Florian P. Limbourg[6] & Ralf H. Adams[1]

While blood vessels play important roles in bone homeostasis and repair, fundamental aspects of vascular function in the skeletal system remain poorly understood. Here we show that the long bone vasculature generates a peculiar flow pattern, which is important for proper angiogenesis. Intravital imaging reveals that vessel growth in murine long bone involves the extension and anastomotic fusion of endothelial buds. Impaired blood flow leads to defective angiogenesis and osteogenesis, and downregulation of Notch signalling in endothelial cells. In aged mice, skeletal blood flow and endothelial Notch activity are also reduced leading to decreased angiogenesis and osteogenesis, which is reverted by genetic reactivation of Notch. Blood flow and angiogenesis in aged mice are also enhanced on administration of bisphosphonate, a class of drugs frequently used for the treatment of osteoporosis. We propose that blood flow and endothelial Notch signalling are key factors controlling ageing processes in the skeletal system.

[1] Faculty of Medicine, Department of Tissue Morphogenesis, Max-Planck-Institute for Molecular Biomedicine and University of Münster, D-48149 Münster, Germany. [2] Research group Integrative Skeletal Physiology, Institute of Clinical Sciences, Imperial College London, Hammersmith Hospital Campus, Du Cane Road, London W12 0NN, UK. [3] Research group Tissue and Tumor Microenvironments, Kennedy Institute of Rheumatology, University of Oxford, Oxford OX3 7LY, UK. [4] Electron Microscopy Unit, Max-Planck-Institute for Molecular Biomedicine, D-48149 Münster, Germany. [5] VIB Vesalius Research Center, KU Leuven, 3000 Leuven, Belgium. [6] Department of Nephrology and Hypertension, Hannover Medical School, D-30625 Hannover, Germany. [7] Department of Plastic and Reconstructive Surgery, Hannover Medical School, D-30625 Hannover, Germany. [8] Research group Ontogeny of Haematopoietic Stem Cells, MRC Centre for Regenerative Medicine, University of Edinburgh, Edinburgh EH16 4UU, Scotland. [9] Molecular Biotechnology Center, Department of Molecular Biotechnology and Health Sciences, University of Torino, 10126 Torino, Italy. Correspondence and requests for materials should be addressed to S.K.R. (email: s.ramasamy@csc.mrc.ac.uk) or to R.H.A. (email: ralf.adams@mpi-muenster.mpg.de).

Osteogenesis is critical for the maintenance of a healthy and fully functional skeletal system. Loss of bone mass is a major health issue associated with ageing and diseases such as osteoporosis. During development of the mammalian skeletal system, bone formation is tightly coupled to angiogenic growth of blood vessels[1–3]. In the embryo, mesenchymal condensations express vascular endothelial growth factor A (VEGF-A), a master regulator of angiogenesis and ligand for the receptor tyrosine kinase VEGFR2 (ref. 4). VEGF-A controls growth plate morphogenesis, cartilage remodelling, blood vessel invasion and ossification during skeletal development[5–7]. Accordingly, bone is a highly vascularized tissue containing an extensive vascular network of large vessels and capillaries. Recently, we have identified a distinct capillary subtype called type H, characterized by high expression of the markers CD31 and Endomucin (Emcn), which couples angiogenesis and osteogenesis in mice[8,9]. Osteoprogenitors, bone forming mesenchymal cells identified by the expression of the transcription factor Osterix, were selectively localized in proximity to type H capillaries but were absent around diaphyseal type L vessels (expressing lower levels of CD31 and Emcn). Type H endothelial cells (ECs) secrete osteogenic factors and maintain Osterix+ osteoprogenitors, but this crucial vessel subtype declined in ageing animals, which was accompanied by reduced osteoprogenitor numbers and loss of bone mass[8,9]. In the bone of ovariectomized mice, a model of osteoporosis, type H capillaries were also reduced[10].

The effect of VEGF-A/VEGFR2 signalling in ECs is strongly linked to the Notch pathway. While VEGF-A promotes EC sprouting and proliferation, these processes are suppressed by Notch receptors and the ligand delta-like 4 (Dll4)[11,12]. Accordingly, reduced Dll4 expression or inhibition of Notch triggered excessive EC sprouting and hyperproliferation in animal models of developmental and tumour angiogenesis[13–16]. Surprisingly, the activation of Notch was found to promote angiogenesis in the bone endothelium, which involved the paracrine (also termed 'angiocrine') release of signals by ECs that are required for chondrocyte maturation, Sox9 expression and VEGF expression[9]. In addition to molecular pathways, the behaviour of ECs is strongly controlled by physical parameters such as blood flow, which has roles in angiogenesis[17], vessel remodelling[18] and numerous vascular pathologies[19,20]. Haemodynamics is also coupled to the homeostasis of the skeletal system[21]. Decreased blood flow was found to be associated with reduced bone mass in elderly women[22]. Similarly, hypertension in older men and women is associated with increased bone mineral density[23]. Case studies reveal that reduced blood supply cause death of bone cells in the osteonecrosis patients[24]. Additionally, active blood supply is essential for callus formation during fracture healing and repair[25]. Impaired blood vessel formation in fractures can result in delayed bone healing and regeneration[26]. Thus, blood flow has been linked to bone repair and maintenance[27], but hardly anything is known about the molecular processes coupling haemodynamics to bone EC function and osteogenesis.

Here, we show that blood flow is crucial for the formation of type H capillaries and angiogenic growth of the vasculature in bone. Disrupted or pharmacologically reduced blood flow results in defective angiogenesis and osteogenesis, and downregulates Notch signalling in bone endothelium. We also find that reduced blood flow and Notch activity in the bone endothelium of aged mice affects angiogenesis and osteogenesis, which is reverted by genetic approaches activating Notch in ECs. The sum of our work highlights central roles of Notch signalling in bone endothelium and its regulation by blood flow, which is relevant for age-related bone loss and, potentially, for therapeutic approaches aiming at the maintenance or restoration of bone mass.

## Results

**Vascular organization and flow pattern in bone**. We investigated the arrangement of arteries, veins and capillaries in tibia to understand fundamental aspects of blood flow pattern in bone. Immunostaining showed that CD31hi α-SMA-covered Emcn-negative arteries and distal arterioles selectively connected to CD31hi Emcnhi capillaries (type H) in the metaphysis and endosteum, but not to diaphyseal sinusoidal (type L) vessels in actively growing (3-week-old) long bone (Fig. 1a, Supplementary Fig. 1a). This pattern indicates that arterial blood enters the long bone at its distal ends as well as at the inner surface of compact bone, and flows through type H capillaries into the highly branched sinusoidal vasculature. From this intricate network, blood is drained by a large Emcn+ CD31lo α-SMA-negative vein with a lumen size of >100 μm in the center of the diaphysis (Fig. 1a–e; Supplementary Fig. 1b). To measure blood velocity in type H and type L vessels of long bone, intravital imaging was performed after dextran injection in living Flk1-GFP reporter mice (Supplementary Movies 1 and 2). Consistent with the hierarchical organization of the bone vasculature, line-scanning analysis in living mice revealed that blood velocity in type H capillaries was much higher ($0.98 \pm 0.1$ mm s$^{-1}$) than in type L sinusoidal vessels ($0.16 \pm 0.04$ mm s$^{-1}$; Fig. 1f). Furthermore, blood velocity dropped further after each vessel branch point in the metaphysis until it finally reached the low velocity characteristic for diaphyseal type L vessels (Fig. 1g). Accordingly, calculated shear rate is significantly higher for type H capillaries than in type L vessels (Supplementary Fig. 1c). The observed differences in blood flow were also in line with the higher transcription of the flow-regulated genes Pecam1, Nos3, Icam1 and Cdh5 in freshly isolated type H ECs than type L endothelium (Supplementary Fig. 1d). These findings raised the possibility that phenotypic differences between type H and L vessels (Fig. 1h) might be linked to blood flow.

**Blood flow regulates bone angiogenesis**. To characterize the role of haemodynamic forces in the generation of type H vessels that mediate angiogenesis, we modulated blood flow in bone. We analysed the 3-week-old tibial vasculature after ligation of the femoral artery (Fig. 2a and Supplementary Fig. 2a), which is the main vessel supplying blood to the lower limb[28,29]. At 48 h post surgery (hps), the inhibition of local blood flow resulted in the strong reduction of columnar vessels and endothelial bud structures in proximity of the growth plate (Fig. 2b and Supplementary Fig. 2b) without promoting endothelial apoptosis (Supplementary Fig. 2c). At an earlier time point after surgery (16 hps), profound anastomoses of buds could be observed in tibial blood vessels near the growth plate (Supplementary Fig. 2d).

Prazosin is an inhibitor of the alpha 1-adrenergic receptor that reduces blood pressure via relaxation of vascular smooth muscle and thereby increases flow into peripheral capillaries[30]. Prazosin has been recently shown to reduce bone formation in mice, which has been linked to alpha 1-adrenoreceptor signalling in osteoblasts[31]. We found that administration of Prazosin for 2 weeks led to significantly reduced blood flow in bone (Fig. 2c), which might result from the diversion of flow into other organs. The observed flow reduction was accompanied by decreased abundance of endothelial buds near the growth plate and of the number of metaphyseal CD31hi type H vessels (Fig. 2d–f) suggesting decreased angiogenesis. Analysis of EdU+ cells on Prazosin treatment showed decreased proliferation of ECs (Fig. 2g and Supplementary Fig. 2e). The expression of flow-regulated genes was significantly downregulated in freshly isolated bone ECs following Prazosin treatment (Supplementary Fig. 2f). Similar phenotypic changes in blood vessels were observed in

mice treated with another flow-reducing drug, Clonidine (Supplementary Fig. 2g,h). Together, these data indicate that angiogenesis in bone is impaired by alterations in blood flow.

**Mode of angiogenesis in bone.** As pharmacological and surgical impairments in blood flow led to profound morphological changes in type H vessels, we investigated the mode of blood

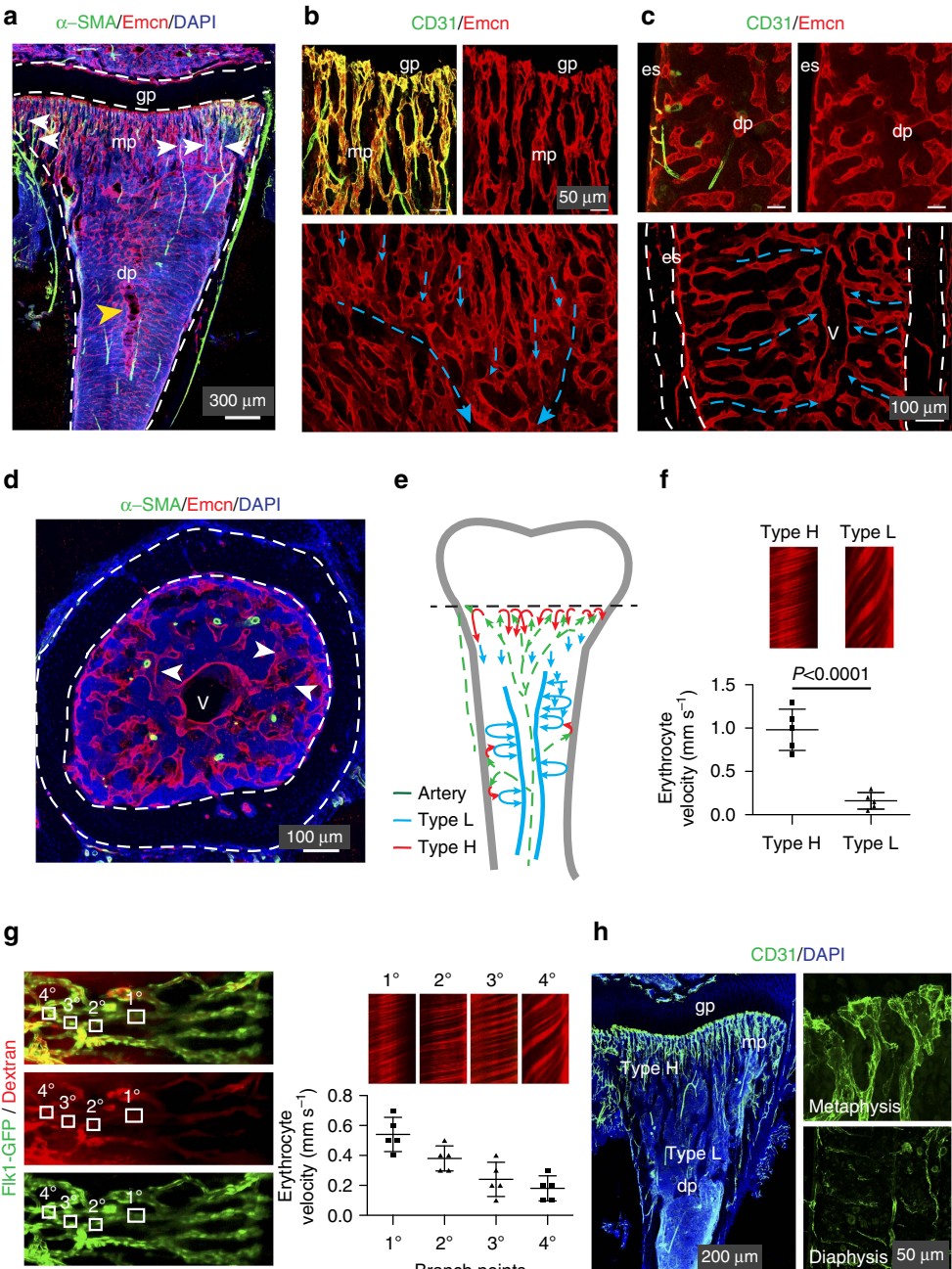

**Figure 1 | Blood flow dynamics in long bone. (a)** Tile scan image of tibial vasculature immunostained for Emcn (red) and α-smooth muscle actin (green). Arteries covered by α-SMA + cells connect (arrows) to metaphyseal (mp) type H vessels near growth plate (dashed lines, gp). Sinusoidal type L vessels connect to central large vein (yellow arrowhead). **(b,c)** Confocal images of 4-week-old metaphysis near growth plate (gp) (**b**, top panels) or diaphysis (dp) (**c**, top). CD31+ (green) Emcn- (red) arteries terminate in CD31+ Emcn+ type H vessels in the metaphysis (mp) and endosteum (es) but not in type L vessels in diaphysis (dp). Blue arrows in Emcn-stained (red) overview images (bottom panels) illustrate blood flow direction from metaphyseal vessel columns (**b**) and endosteum (**c**) into the adjacent sinusoidal network and veins. **(d)** Maximum intensity projection of transversal sections of 4-week-old tibia immunostained for α-SMA (green) and Emcn (red). Sinusoidal type L vessels (arrowheads) connect to a large central vein (v). Multiple smooth muscle-covered CD31+ Emcn-arteries cross the diaphysis. Dashed lines mark compact bone. **(e)** Diagram of arterial (green arrows), type H (red arrows) and sinusoidal/venous flow (blue arrows) in murine long bone. **(f)** Graph showing blood velocities calculated from line scanning of type H and type L vessels after Dextran (2,000,000 Da) injection. Data represent mean ± s.d. ($n = 5$ biological replicates). P values, two-tailed unpaired t test. **(g)** Overview image and representative line scans showing blood velocities in 2-week-old long bone. Note decreasing velocity of erythrocytes at each branch point (represented as 1°, 2°, 3°, 4°) at the interface between columnar type H (right) and diaphyseal type L vessels (left). Data represent mean ± s.d. ($n = 5$ biological replicates). **(h)** Confocal images of CD31 (green) expression in the metaphysis and diaphysis of 4-week-old tibia. Note differences in CD31 levels. Scale bars are as indicated in the respective images. DAPI (blue) is used for counterstaining of nuclei.

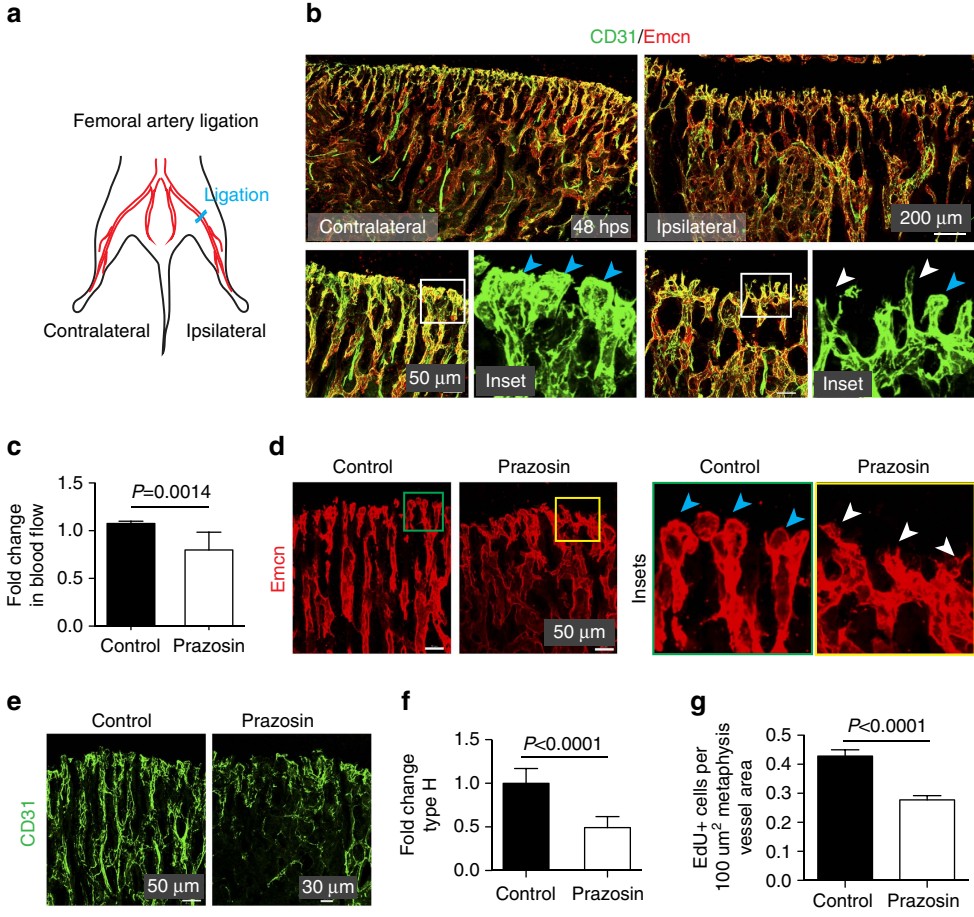

**Figure 2 | Changes in blood flow impair bone angiogenesis.** (**a**) Schematic representation of unilateral femoral artery ligation in 3-week-old mice. Results compare operated (ipsilateral) legs with contralateral controls. (**b**) Representative confocal images showing CD31 (green) and Emcn (red) immunostaining of tibia from ligated and contralateral side at 48 h post surgery (hps). Note decline in vascular front structures, buds and columns on ligation. Insets show vessel buds (blue arrowheads) on the contralateral side and appearance of pointed, sprout-like structures (white arrowheads) after ligation. (**c**) Analysis of blood flow in Prazosin-treated tibia compared with saline-treated control. Data represent mean ± s.d. ($n = 4$ biological replicates). $P$ values, two-tailed unpaired $t$ test. (**d,e**) Confocal images of 4-week-old Prazosin-treated and control tibial metaphysis immunostained for Emcn (red; **d**) or CD31 (green; **e**). Insets in **d** show vessel buds (blue arrowheads), which appeared collapsed (white arrowheads) after Prazosin treatment. CD31 (green) expression was decreased in Prazosin-treated animals (**e**). (**f**) Flow cytometric quantification of tibial type H ECs after treatment with Prazosin or control (saline). (**g**) Quantitation of EdU + ECs present in the metaphysis of Prazosin-treated and control mice. Data represent mean ± s.e.m. ($n = 4$ biological replicates). $P$ values, two-tailed unpaired $t$ test.

vessel growth in postnatal bone. While the columnar organization is a characteristic feature of metaphyseal type H vessels, these columns were interconnected by distal, loop-like arches. Bud-shaped protrusions emerging from those arches were localized in close proximity of hypertrophic growth plate chondrocytes (Fig. 3a). Their juxtaposition to hypertrophic chondrocytes, a known source of VEGF-A (ref. 5), suggested that buds represent the leading edge of the growing vasculature, which would make them functionally equivalent to endothelial tip cells and sprouts observed in other organs[32,33]. Accordingly, the length of columns and the abundance of buds increased during postnatal bone growth (Supplementary Fig. 3a–c). In adult (12-week-old) long bone, arches remained as the most prominent distal vessel structure, while the abundance of buds was strongly decreased (Fig. 3b; Supplementary Fig. 3b). In line with the previously reported reduction of type H vasculature in long bone of aged mice[8], CD31[hi] vessel arches in 80-week-old mice were sparse and spanned longer distances than in adults, while buds were almost completely absent (Fig. 3b).

To gain insight into the dynamics of blood vessel formation in postnatal long bone, we developed an intravital live imaging technique using the endothelium-specific *Flk1-GFP* reporter mice to monitor blood vessel growth in long bone. The observation of green fluorescent protein (GFP) + ECs next to the growth plate demonstrated the emergence of buds from arches (Fig. 3c and Supplementary Movie 3). Static imaging and ultrastructural analysis by scanning electron microscopy indicated that lumenized buds carried short filopodia, which connected to the surrounding chondrocyte matrix (Fig. 3d,e and Supplementary Fig. 4a). Other buds lacked visible filopodia and were found in proximity of intact chondrocytes (Fig. 3e and Supplementary Fig. 4b–d), suggesting that they were not actively extending at the time of analysis. Mosaic fluorescent labelling of ECs by tamoxifen-induced, EC-specific recombination of *R26-Confetti* reporter mice indicated that buds were composed of multiple cells enclosing a luminal space (Fig. 3f and Supplementary Fig. 4e,f). Dynamic and static imaging approaches also established that new arches were formed through the anastomoses of two neighbouring bud structures (Fig. 3g–i and Supplementary Movie 4). Thus, arches and buds are dynamic structures that are formed during active angiogenesis and mediate vessel growth in long bone. Due to reduced or absent endothelial budding after

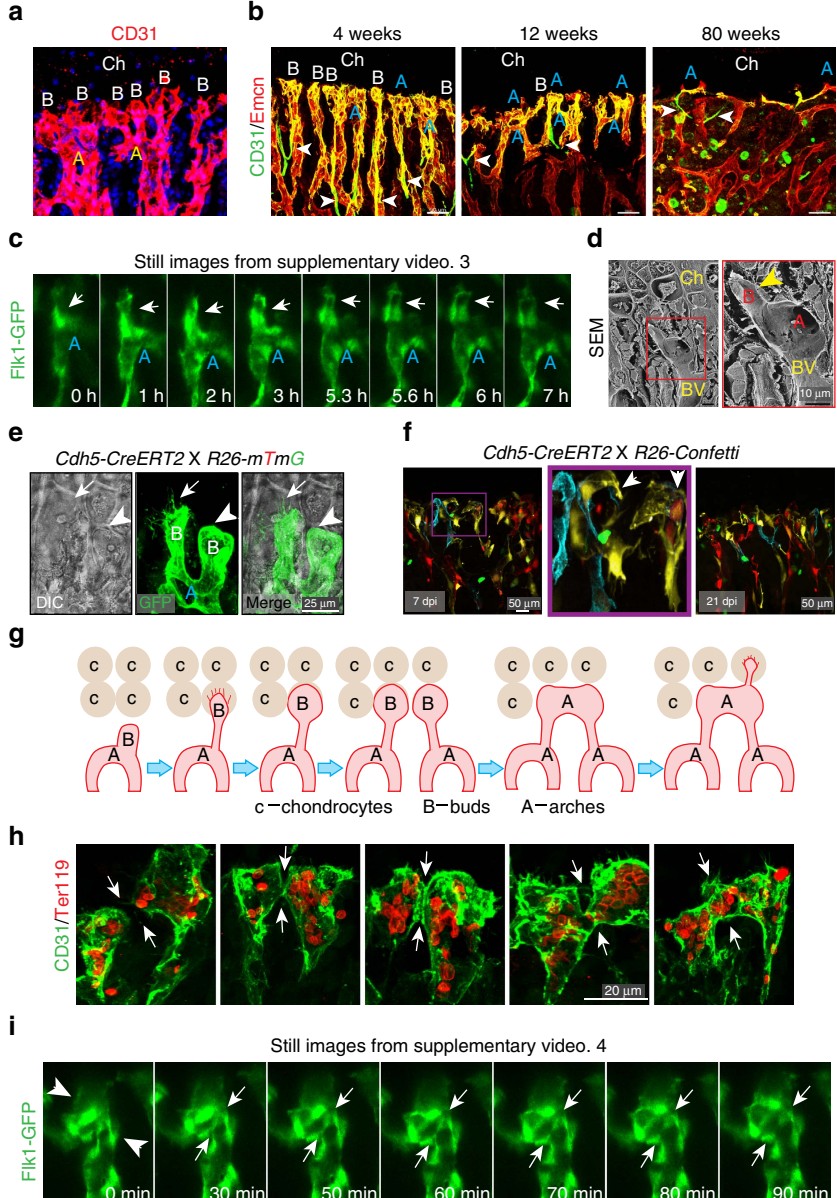

**Figure 3 | Blood vessel growth in bone.** (**a**) Maximum intensity projection of metaphyseal vessel structures (CD31, red): distal, loop-like arches (A) and bud-shaped protrusions (B) present in proximity of hypertrophic chondrocytes (Ch). (**b**) Representative confocal images of tibia sections from 4, 12 and 80-week-old mice immunostained for Emcn (red) and CD31 (green). CD31[hi] Emcn[hi] ECs form columns, distal arches (A) and buds (B) at the vascular growth front, which were abundant at 4 weeks but declined during ageing (12 and 80 weeks). Arrowheads indicate Emcn- arterioles. Ch, chondrocytes. (**c**) Still images of the indicated time points from a 10 h time lapse movie of 10-day-old *Flk1-GFP* metatarsal. Arrow indicates bud emerging from arch (A). (**d**) SEM images of blood vessel (BV) next to chondrocytes (Ch) in 4-week-old tibia with distal arch (A) and bud (B; arrowhead). Right panel shows higher magnification of inset with bud emerging from arch. (**e**) Merged confocal and differential interference contrast (DIC) image showing distal vessel buds (B) genetically labelled by GFP (green). Arrow indicates a filopodia-bearing endothelial bud displacing an apoptotic chondrocyte, arrowhead points at filopodia-free bud next to intact chondrocyte. (**f**) 4-week-old *Cdh5-CreERT2 R26-Confetti* double transgenics at 7 and 21 days post injection (dpi) showing high degree of mosaicism in endothelial columns, arches and buds. (**g**) Schematic illustration of vessel growth in long bone. Invading vessels buds (B) displace apoptotic growth plate chondrocytes (c) and, through anastomosis, generate new arches (A), from which new buds can emerge. (**h**) Representative confocal images showing contact formation and anastomosis (arrows) of CD31+ (green) vessel buds in 4-week-old tibia. Note red presence of Ter-119+ RBCs (red) in buds and forming vessel arches. (**i**) Still images (time indicated) from movie showing anastomosis (arrows) of 2 buds (arrowheads in left image) in 10-day-old *Flk1-GFP* (green) metatarsal. Arrows show the disappearance of junction between anastomosing buds at later time points.

the completion of postnatal bone growth, vessel arches remain as the predominant distal vascular structure in adult and aged long bone (Fig. 3b). These results also indicate that the defective angiogenesis observed under flow-modulated conditions is due to defective bud formation and anastomosis at the vascular front (Fig. 2b and Supplementary Figs 2b and 4g).

**Flow-mediated coupling of angiogenesis and osteogenesis.** As angiogenesis in bone is tightly coupled with osteogenesis, we investigated whether modulation of blood flow would affect bone formation. Both femoral artery ligation and Prazosin treatment led to significant reductions in Osterix-expressing osteoprogenitors (Fig. 4a–c and Supplementary Fig. 5a,b).

Moreover, micro-CT analysis confirmed that Prazosin administration led to loss of mineralized bone (Fig. 4d,e and Supplementary Fig. 5c). This is likely to reflect altered bone formation rates, as osteoclast numbers were not significantly changed (Supplementary Fig. 5d). Defective blood flow also led to reduced expression of pro-osteogenic factors in freshly sorted bone ECs (Fig. 4f).

Next, we investigated the molecular link between blood flow and angiogenesis in bone. Pecam1/CD31 is highly expressed by ECs in type H arch and bud structures (Fig. 1h) and was shown to have mechanosensory activity[34,35]. To evaluate a potential involvement of CD31 in the generation of specific vessel subtypes, we analysed the vasculature in long bone of *Pecam1* knockout mice[36]. However, in spite of absent CD31 expression, these mutants did not show defects in bone angiogenesis

(Supplementary Fig. 6a). Likewise, Osterix-expressing cells were unaffected in Pecam1-deficient mice (Supplementary Fig. 6b) indicating that this molecule is dispensable for physiological bone angiogenesis and osteogenesis.

**Flow modulates endothelial Notch signalling.** Notch signalling in bone endothelium promotes blood vessel growth and has been shown to couple angiogenesis and osteogenesis[9]. Immunostaining showed that Dll4, the critical Notch ligand in bone ECs, was highly expressed on the endothelium of arteries and distal type H buds, which was independent of PECAM1/CD31 expression (Supplementary Fig. 6c). The transcription factor RBP-Jκ is an essential regulator of Notch-induced gene expression inside and outside the vasculature[37–39], and *Cdh5(PAC)-CreERT2*-mediated

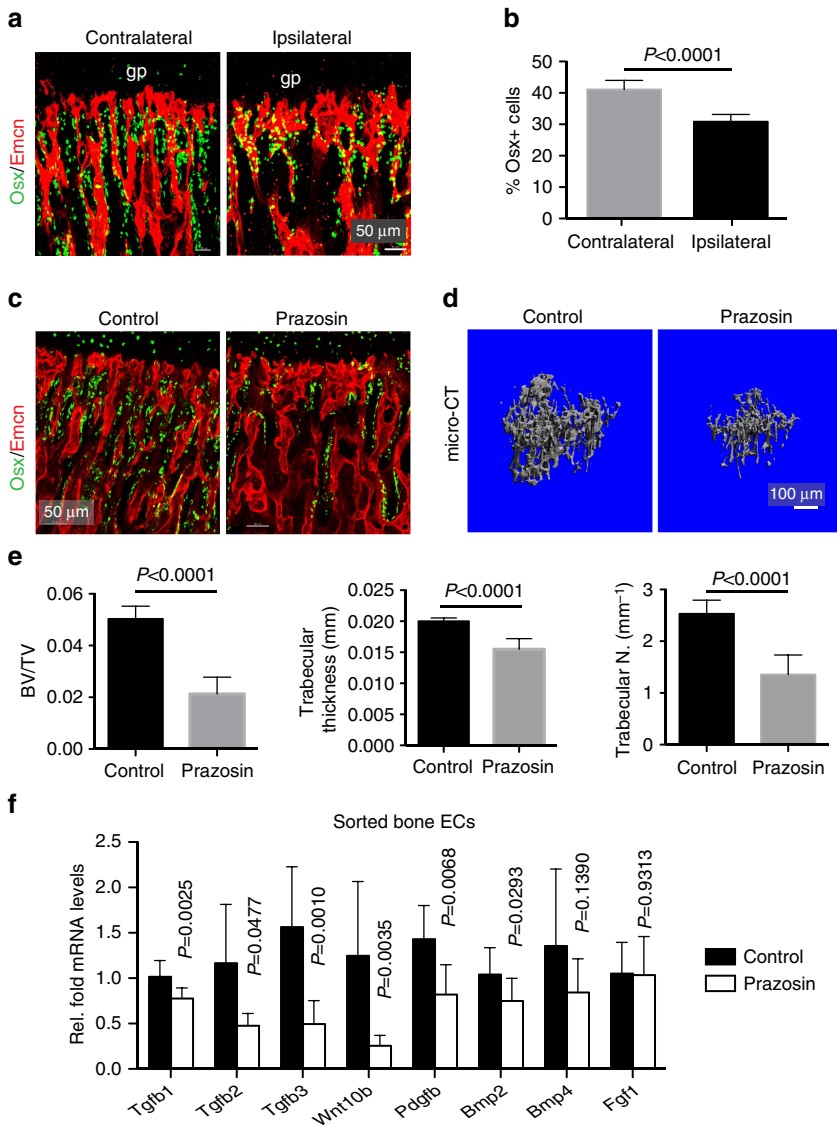

**Figure 4 | Flow-mediated angiogenesis is coupled to osteogenesis.** (**a**) Maximum intensity projections of Emcn (red) and Osterix (Osx, green) immunostaining in contralateral (control) and ipsilateral (ligated) tibia at 48 hps after femoral artery ligation. (**b**) Quantitation of Osx + cells in contralateral and ipsilateral tibia after femoral artery ligation. Data represent mean ± s.e.m (n = 5 mice in three independent experiments). P values, two-tailed unpaired t test. (**c**) Confocal images of Prazosin and control tibial sections immunostained for Emcn (red) and Osterix (Osx, green). (**d**) 3D rendering showing micro-CT analysis of mineralized regions in the metaphysis of control and Prazosin-treated tibiae. (**e**) Histomorphometrical data derived from micro-CT scans. Note reduced bone parameters after Prazosin treatment. (**f**) RT-qPCR for the expression of known pro-osteogenic factors such as *Tgfb1*, *Tgfb2*, *Tgfb3*, *Wnt10b*, *Pdgfb*, *Bmp2*, *Bmp4* and *Fgf1* (normalized to *Actb* expression) in sorted bone ECs from control and Prazosin-treated mice. Data represent mean ± s.d. (n = 8 biological replicates). P value, two-tailed unpaired t test.

inactivation of the *Rbpj* gene led to defective growth and severe dilation of metaphyseal vessels in the resulting *Rbpj*[iΔEC] mutants[9]. Consistent with these morphological alterations, transcript analysis of sorted *Rbpj*[iΔEC] bone ECs revealed a downregulation of the flow-modulated genes *Klf2*, *Nos3* and *Pecam1* relative to control littermates (Supplementary Fig. 7a). Conversely, EC-specific and inducible (*Cdh5(PAC)-CreERT2*-controlled) overexpression of the active Notch1 intracellular domain (NICD), which was recently shown to enhance artery formation in long bone[40], led to the upregulation of *Klf2*, *Nos3* and *Pecam1* in ECs (Supplementary Fig. 7a). Thus, Notch signalling controls the expression of flow-controlled genes, which presumably reflects morphological alterations in the bone vasculature.

Other results indicate that the expression of Notch pathway components and Notch signalling in ECs are regulated by flow. Prazosin treatment led to a strong reduction of Dll4 expression by bone ECs at the transcript and protein level (Fig. 5a,b). Likewise, mRNAs for several Notch pathway target genes were significantly downregulated in freshly isolated bone ECs from Prazosin-treated animals (Fig. 5b). Tibial ECs from femoral artery ligation experiments (16 hps) also showed reduced expression of Dll4 (Supplementary Fig. 7b) and other Notch downstream target genes (Fig. 5c). Conversely, when mice were treated for 48 h with angiotensin II to increase blood flow, sorted bone EC showed increased expression of Notch target genes (Supplementary Fig. 7c). The exposure of cultured ECs to laminar flow increased the level of NICD protein and upregulated the expression of known Notch target genes (Supplementary Fig. 7d,e). Together, these data indicate that endothelial Notch signalling is controlled by blood flow.

**Blood flow in aged mice bones**. Skeletal ageing is associated with decreased bone formation and loss of type H vasculature[8]. Gene expression analysis in freshly isolated bone ECs from aged mice revealed that transcript levels for numerous Notch target genes were strongly downregulated (Fig. 5d,e). Measurements using radiolabelled microspheres in rats have previously shown that blood flow in aged bone is decreased relative to young adult animals[41]. Our own analysis also indicated a remarkable decline in blood flow in long bone of aged mice relative to young animals (Fig. 5f). These data suggest that the reductions in both Dll4 expression and Notch signalling in ECs of the aged bone vasculature might be induced or enhanced by changes in blood flow. To further investigate the link between endothelial Notch activity, blood flow and organization of the bone vasculature, Prazosin was administered to transgenic mice overexpressing active NICD of Notch1 (*NICD*[iOE-EC])[42] in ECs. Remarkably, bud structures at the vascular growth front were preserved in Proazosin-treated *NICD*[iOE-EC] gain-of-function mutants (Fig. 5g and Supplementary Fig. 7f). Thus, Notch activation in bone ECs was sufficient to overcome defects caused by administration of Prazosin.

Next, we addressed whether EC-specific Notch activation would be also sufficient to overcome age-related changes in the bone vasculature. For this purpose, *NICD*[iOE-EC] mutants at the age of 55–65 weeks received injections of tamoxifen before analysis 42 days later (Fig. 5h). This approach led to the reappearance of type H vessels and formation of Dll4-positive endothelial buds in the *NICD*[iOE-EC] but not in the control metaphyseal region (Fig. 5i,j and Supplementary Fig. 8a,b). A second strategy involved EC-specific inactivation of the gene encoding Fbxw7 (*Fbxw7*[iΔEC])[43], which mediates the polyubiquitination and proteasomal degradation of active Notch. Tamoxifen treatment of aged *Fbxw7*[iΔEC] mice also lead to the reappearance of CD31[hi] capillaries together with

Dll4-positive vessels buds, which were absent in control littermates (Fig. 5h and Supplementary Fig. 8c). Changes in the vasculature of aged *NICD*[iOE-EC] and *Fbxw7*[iΔEC] mutants were accompanied by significant increases in Osterix-expressing osteoprogenitors (Fig. 5k; Supplementary Fig. 8d,e) and enhanced bone formation, as seen by micro-CT analysis (Supplementary Fig. 9a–d). The formation of new bone also involved enhanced bone remodelling indicated by the increased abundance of osteoclasts (Supplementary Fig. 9e). Thus, EC-specific Notch-mediated reactivation of type H vessels in aged mice can promote both angiogenesis and osteogenesis leading to improved bone mass.

**Bisphosphonate increases bone blood flow and angiogenesis**. Bisphosphonates are commonly used medications for osteoporosis treatment. While the inhibition of hydroxyapatite breakdown by osteoclasts is the main mechanism of action for bisphosphonates, it has been proposed that this class of antiresorptive drugs also acts on other cell types and inhibits apoptosis of osteoblasts and osteocytes[44,45]. As age-related changes in bone mass in mice strongly correlate with alterations in the abundance of CD31[hi] vessels in bone[8], blood flow, EC proliferation and the presence of endothelial buds (Fig. 6a), we tested whether bisphosphonate treatment would affect any of these vascular parameters. Indeed, administration of Alendronate to aged mice (2 mg kg$^{-1}$) twice a week for 6 weeks led to substantial increases in type H vessels and Osterix + osteoprogenitors relative to control animals (Fig. 6b,c). Alendronate also induced the reappearance of vessel buds in the metaphysis and expression of the Notch ligand Dll4 (Fig. 6d). Expression analysis of freshly sorted ECs from aged bone showed that Alendronate significantly increased transcript levels of several Notch target genes (Fig. 6e), whereas such changes were not induced by Alendronate in cultured primary bone ECs (Supplementary Fig. 9f). In addition, Alendronate treatment led to significantly increased blood flow (Fig. 6f). To understand whether the increase in blood flow on Alendronate treatment precedes or succeeds new blood vessel formation, we measured flow in the bone vasculature after short treatment with bisphosphonate. Following three injections of Alendronate every 12 h, 3-week-old mice showed significantly increased perfusion of the bone vasculature at 30 min after last injection (Fig. 6g). Taken together, these data argue that bisphosphonate treatment can trigger a complex response in bone, which involves enhancements in blood flow, activation of endothelial Notch signalling and vessel growth.

## Discussion
In addition to their conventional role as a conduit system for blood circulation, there is increasing evidence that vascular cells generate specialized microenvironments supporting stem and progenitor cells. In bone, the local vasculature provides niches for haematopoietic stem cells and thereby regulates haematopoiesis[46–49]. Likewise, bone formation and fracture healing are controlled by blood vessels and EC-derived molecular signals[8,9,50,51]. The osteogenesis-promoting effect of the bone vasculature in physiological settings was recently attributed to the type H capillary subset, which is characterized by high expression of the cell adhesion protein CD31/PECAM1 and association with perivascular Osterix-positive cells[8,9]. Indicating potential roles in disease and therapy, CD31[hi] Emcn[hi] capillaries were decreased in ovariectomy-induced osteoporotic mice, whereas the administration of cathepsin K inhibitor, which impairs bone resorption, led to an increase in CD31[hi] Emcn[hi] vasculature[10].

Bone growth is controlled by genetic programs and physical factors such as mechanical loading[52,53]. Chondrogenesis, chondrocyte differentiation, VEGF expression in the growth plate and angiogenesis are also influenced by mechanical signals[5,54,55]. Our current study unravels the cellular basis of angiogenic vessel growth in the metaphysis by type H capillaries in direct proximity of growth plate chondrocytes with live imaging, which revealed that new anastomotic connections (arches) are formed by fusion of multicellular vessel buds. Blood flow is required for the formation and anastomotic fusion

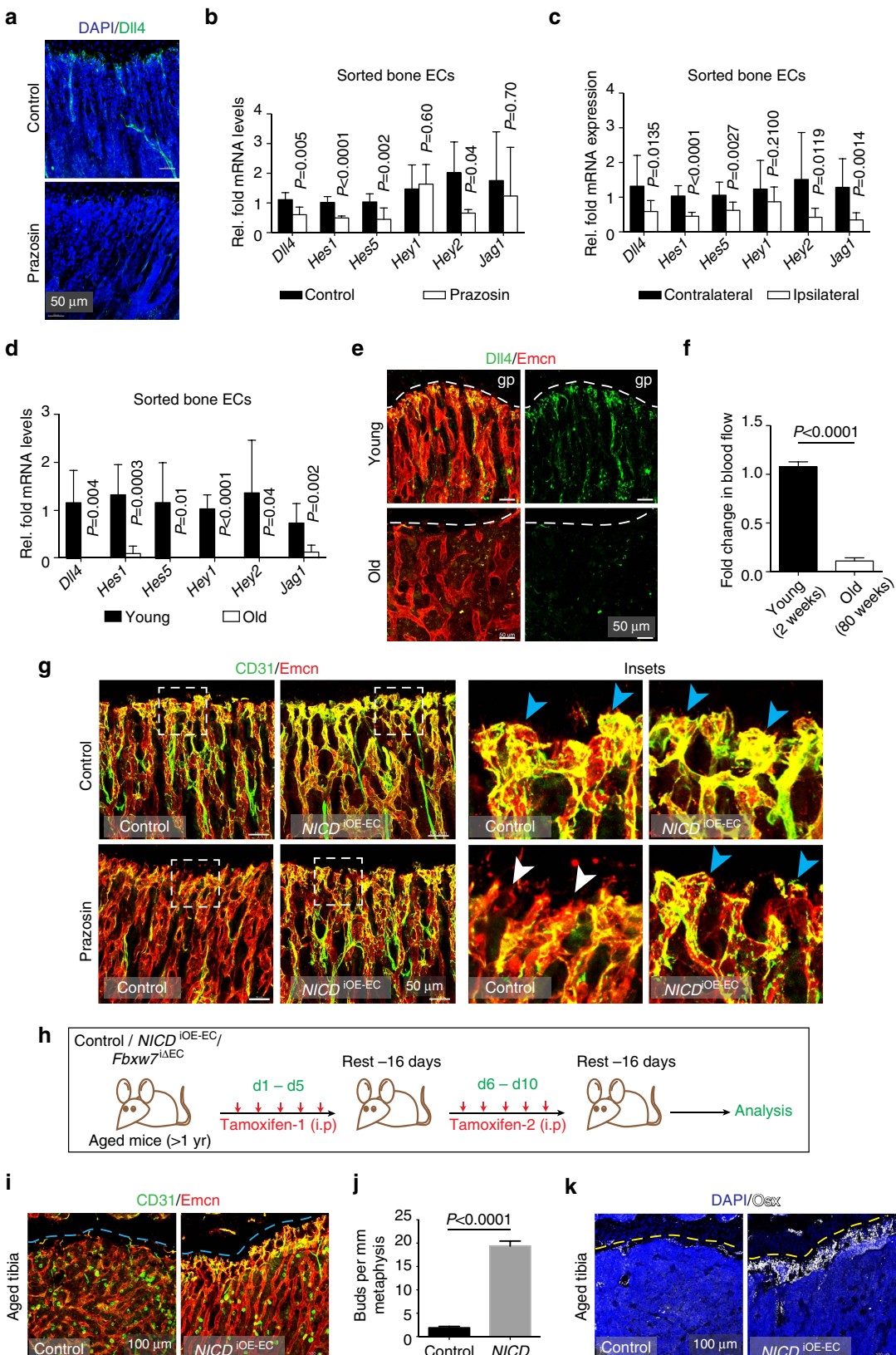

of buds in the vascular front, type H vessel formation and angiogenesis in bone. Thus, CD31[hi] Emcn[hi] capillaries are not only connecting directly to distal arterioles and are therefore perfused at higher velocity than the highly branched sinusoidal network, but blood flow also regulates fundamental properties of the bone vasculature and its ability to promote osteogenesis. The relationship between blood pressure, flow in the skeletal system and bone mass in the clinical context is complex and linked to numerous factors such as nutrient delivery, the regulation of local metabolism or the influx of calcium and phosphate[21,27]. Decreased blood flow in bone has been reported in pathological conditions such as osteoporosis[22] and osteonecrosis[24]. Likewise, ageing is not only associated with bone loss but also with reduced blood flow in the skeletal system[41]. Our new findings link these flow and ageing-related changes to reduced Notch activity in ECs. While endothelial Notch signalling is known to suppress vessel growth in a wide range of organs and model organisms[13,56,57], the pathway acts as a strong stimulator of angiogenesis and thereby osteogenesis in the skeletal system[9]. The expression of Notch pathway genes and bone formation were strongly reduced by treatments disturbing normal blood flow or in aged animals, whereas the reactivation of Notch signalling in ECs was sufficient to restore local angiogenesis and bone formation (Fig. 7).

A number of reports addressing the relationship between flow and Notch signalling suggest complex and, potentially, context-dependent interactions. Notch pathway genes are highly expressed by arterial ECs, which are exposed to higher flow rates and shear stress, whereas venous expression is low or undetectable[58,59]. This is consistent with numerous studies reporting Notch activation in blood vessel ECs in response to flow and laminar shear stress[60–62]. In contrast, loss of blood flow led to increased vascular Notch signalling in the vasculature of zebrafish embryos[63]. Notch facilitates vessel constriction during the regression (pruning) of capillaries[64], but Notch activity is also required for nitric oxide induction and vasodilation in response to ischaemia in the adult vasculature[65]. Furthermore, Notch activation is a hallmark of arteriovenous malformations and constitutive activation of Notch4 has been shown to induce the enlargement of capillary vessels into arteriovenous shunts[66]. We propose that Notch controls blood flow and expression of flow-controlled genes in the bone endothelium presumably by controlling vascular morphology, angiogenic growth and the formation of arteries/arterioles in bone. Whether such effects might also involve the release of vascular autocoids (that is, local acting factors) such as nitric oxide, which was shown to reduce bone resorption[67], or pro-anabolic molecules that promote bone formation, such as calcitonin gene-related peptide[68], remains to be investigated. Notch signalling also has important cell-autonomous roles in the osteoblast lineage and preserves mesenchymal progenitors by preventing excessive differentiation of these cells[69,70]. Notch activation in mature osteocytes triggers a strong skeletal anabolic response in adult mice, which is sufficient to rescue age-associated and ovariectomy-induced bone loss[71]. As osteocyte processes physically interact with the vascular endothelium[72], it would be worthwhile to investigate whether different cell types communicate through direct Notch-ligand interactions.

Our findings also support that Notch activity and pathway gene expression in ECs depend on flow, which is therefore critical for the formation of type H capillaries, the formation of endothelial buds and thereby the expansion of the growing vasculature, and Notch-mediated coupling of angiogenesis and osteogenesis. Accordingly, reduced blood flow in the ageing organism might be a cause of reduced endothelial Notch activity, which, in turn, might contribute to age-related bone loss. Supporting this concept, the reactivation of Notch signalling in the ageing vasculature was sufficient to induce local growth of type H capillaries, which was accompanied by the expansion of vessel-associated Osterix + osteoprogenitors and formation of mineralized bone.

Remarkably, treatment of mice with bisphosphonate, the class of drugs that is commonly used in humans to prevent the loss of bone mass, also led to the induction of endothelial Notch expression, the emergence of CD31[hi] Emcn[hi] capillaries and increased blood flow. While it remains unclear whether the effect of Alendronate on bone ECs is direct or mediated by changes in other cell types, our findings nevertheless indicate that properties of the local vasculature might play a far more central role in the regulation of bone mass than previously appreciated.

## Methods

**Age groups and genetically modified mice.** C57BL/6 J males were used for all experiments involving wild-type mice and pharmacological treatments, whereas both males and females animals were analysed in the genetic experiments. Mice at the age of 2–4 weeks and > 53 weeks were chosen for young and aged groups, respectively, unless specified otherwise. For Prazosin treatment, C57BL/6 J mice were given 0.5 mg kg$^{-1}$ body weight dose every second day from P10 to P20 before they were killed and analysed at P24. For Alendronate treatment in aged mice, 75-week-old C57BL/6 J mice were given 2 mg kg$^{-1}$ body weight dose twice a week for 5 weeks and killed on the 6th week. Mosaic multi-colour labelling of ECs was achieved by combining *R26R-Confetti* reporter[73], which can express four different fluorescent proteins in a stochastic and Cre-dependent fashion, with *Cdh5(PAC)-CreERT2* transgenics[8,74] expressing tamoxifen-inducible CreERT2 specifically in ECs. 4-Hydroxy tamoxifen was administered (100 μg) by five consecutive, daily injections, which was followed by analysis of the bone vasculature at 7 or 21 days after the last injection, as indicated. For *Fbxw7* deletion in the vasculature of aged mice, mice carrying loxP-flanked *Fbxw7* gene (*Fbxw7* [floxed])[43] were interbred with the *Cdh5(PAC)-CreERT2* line. *Fbxw7*[floxed/floxed] *Cdh5(PAC)-CreERT2* [T/+] males were interbred with *Fbxw7*[floxed/floxed] females to generate litters with *Fbxw7*[floxed/floxed] *Cdh5(PAC)-CreERT2* [T/+] (*Fbxw7*[iΔEC]) mutants and Cre negative *Fbxw7*[floxed/floxed] (control) littermates. To induce Cre-mediated gene inactivation,

**Figure 5 | Flow positively regulates endothelial Notch signalling in bone.** (**a**) Dll4 (green) immunostaining of Prazosin-treated and control tibia sections. (**b,c**) qPCR analysis of *Dll4, Hes1, Hes5, Hey1, Hey2* and *Jag1* expression (normalized to *Actb*) in ECs sorted from Prazosin-treated or control long bone (**b**) and from ipsilateral (operated) and contralateral sides of femoral artery ligated (16hps) limbs of mice (**c**). Data represent mean ± s.d. (n = 5 biological replicates). P values, two-tailed unpaired t test. (**d**) qPCR analysis of *Dll4, Hes1, Hes5, Hey1, Hey2* and *Jag1* expression (normalized to *Actb*) in ECs sorted from 4 (young) and 85-week-old (old) long bone. Data represent mean ± s.d. (n = 5 biological replicates). P values, two-tailed unpaired t test. (**e**) Representative confocal images showing the metaphysis region near the growth plate (gp, dashed line) in young (4 week) and aged (85 week) tibia immunostained for Emcn (red) and Dll4 (green). (**f**) Blood flow measurements in young and old mice show reduced flow upon ageing. Data represent mean ± s.d. (n = 5 biological replicates). P values, two-tailed unpaired t test. (**g**) Maximum intensity projections of tibial sections from *NICD*[iOE-EC] (Notch gain-of-function) mice and littermate controls treated with saline (control) and Prazosin, immunostained for CD31 (green) and Emcn (red). Blue arrowheads indicate buds and arches, white arrowheads mark defective buds in the vascular front. (**h**) Experimental scheme of tamoxifen administration to aged transgenic mice for EC-specific activation of Notch signalling. (**i**) Confocal images of aged *NICD*[iOE-EC] and littermate control tibiae immunostained for CD31 (green) and Emcn (red). Note increase in CD31 + vessels and buds (arrows) near *NICD*[iOE-EC] growth plate (gp, dashed blue line). (**j**) Quantification of bud structures in the vascular front of aged *NICD*[iOE-EC] and littermate control tibiae. Data represent mean ± s.d. n = 4 in three independent experiments. P values, two-tailed unpaired t test. (**k**) Confocal images of the metaphyseal region of aged *NICD*[iOE-EC] and littermate control tibiae immunostained for Osterix (Osx, white). Nuclei, DAPI (blue).

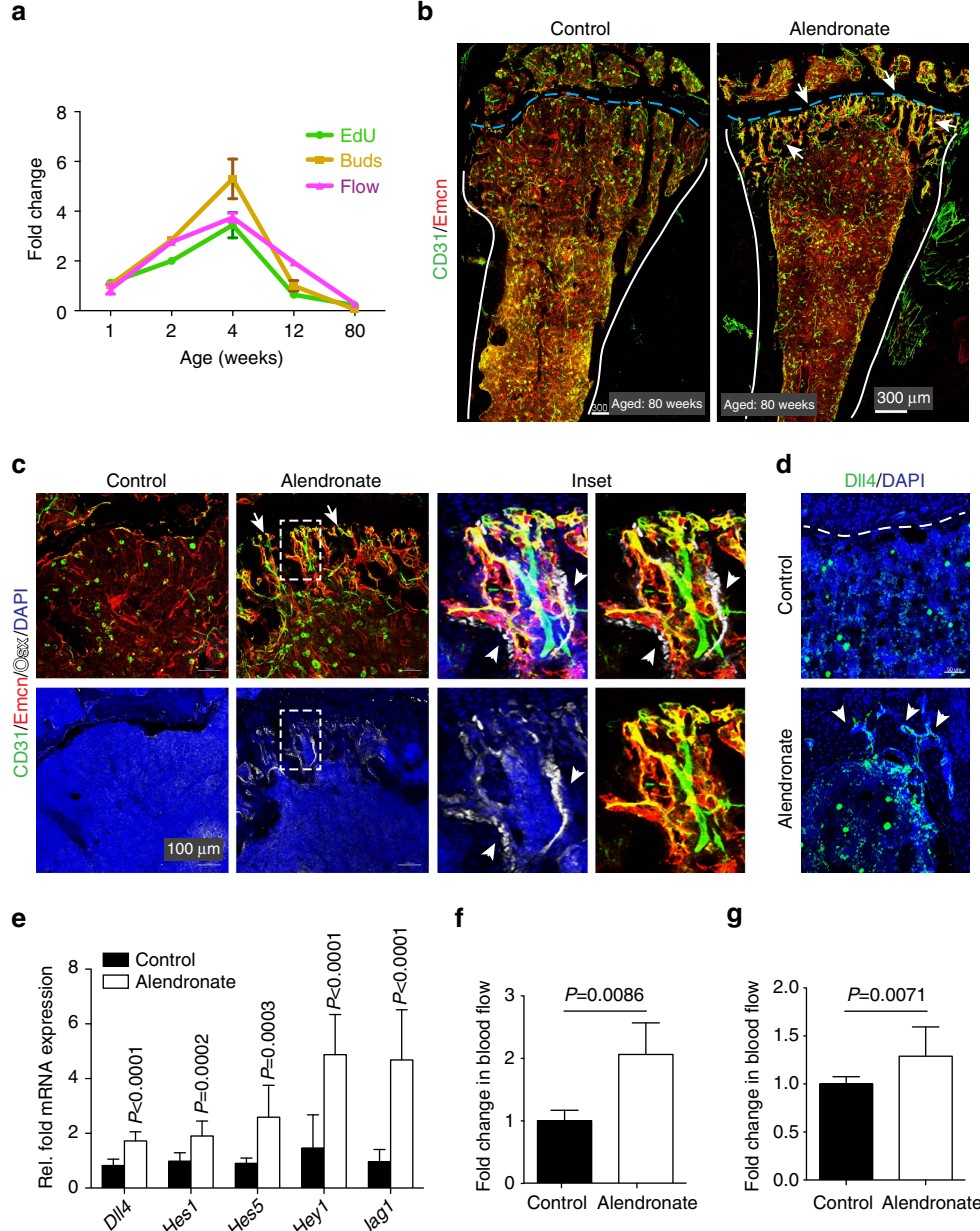

**Figure 6 | Alendronate stimulates endothelial Notch signalling and flow in bone. (a)** Graph showing age-dependent changes in EdU+ ECs and bud structures in distal tibia. Blood flow measurements in the same age groups indicate a similar pattern. Data represent mean ± s.d. ($n = 5$ biological replicates). $P$ values, two-tailed unpaired $t$ test. **(b)** Tile scan images of tibia from control and Alendronate-treated 80-week-old mice. Sections were immunostained for CD31 (green) and Emcn (red). Note appearance of CD31+ blood vessels (arrows) in Alendronate-treated metaphysis. **(c)** Vascular front in sections of control and Alendronate-treated aged tibia immunostained for CD31 (green), Emcn (red) and Osterix (Osx, white). Nuclei, DAPI (blue). Inset shows CD31+ blood vessels in close proximity to osteoblasts. **(d)** Confocal images showing metaphysis of aged mice after Alendronate administration. Immunostaining shows endothelial Dll4 (green). Nuclei, DAPI (blue). **(e)** qPCR analysis of *Dll4, Hes1, Hes5, Hey1* and *Jag1* transcripts expression in ECs sorted from control and Alendronate-treated 80-week-old long bone. Data represent mean ± s.d. ($n = 6$ biological replicates). $P$ values, two-tailed unpaired $t$ test. **(f)** Increased blood flow in tibia of Alendronate-treated aged (80-week-old) mice. Data represent mean ± s.d. ($n = 6$ biological replicates). $P$ values, two-tailed unpaired $t$ test. **(g)** Blood flow measurements in tibia after short-term treatment with Alendronate. Data represent mean ± s.d. ($n = 6$ biological replicates). $P$ values, two-tailed unpaired $t$ test.

aged (>1 year) mutant and control mice were injected with 1,000 μg tamoxifen (Sigma, T5648) intraperitoneally every day for 5 days. All mice were rested for 16 days before subjecting them to a second round of 5 tamoxifen injections. After another 16-day resting period, mice were analysed by collecting femur and tibia after euthanasia. For EC-specific overexpression of the active NICD, *Gt(ROSA)26Sor*$^{tm1(Notch1)Dam/J}$ mice[42] carrying a Cre-inducible transgene for NICD overexpression and *Cdh5(PAC)-CreERT2* transgenics were interbred. Tamoxifen administration (see injection schedule for aged mice above) was used to generate *CreERT2*-positive (*NICD*$^{iOE-EC}$) Notch gain-of-function mutants and corresponding Cre-negative control littermates.

Experiments involving animals were performed following protocols approved by local animal welfare boards and official institutions (LAVES Lower Saxony, LANUV North Rhine Westphalia).

**Femoral artery ligation.** Male 3-week-old C57BL/6 J mice were subjected to permanent femoral artery ligation[28]. Mice were anaesthetized by intraperitoneal injection of ketamine and xylazine before surgery. The femoral artery was ligated distal to the origin of the deep femoral artery and proximal to the popliteal artery. Blood flow measurements in mouse feet were performed on 37 °C heated pads before and immediately after surgery and on post-operative 16 and 48 h using a

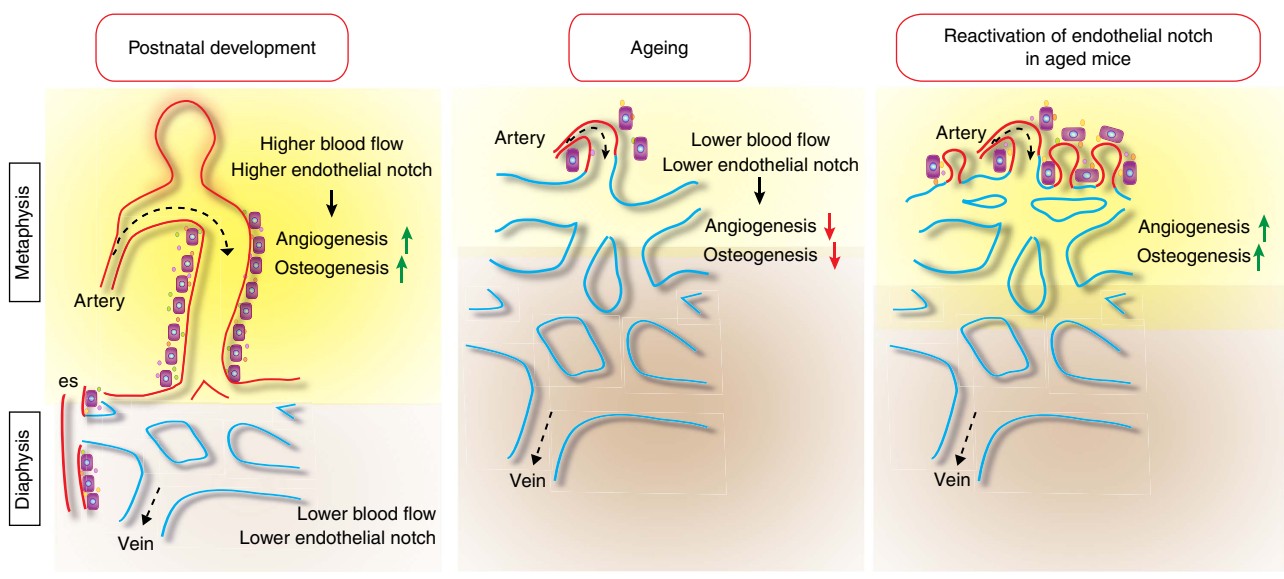

**Figure 7 | Schematic representation of key findings.** Postnatal angiogenesis in bone involves the formation of arch and bud structures. Blood flow is higher in metaphyseal type H vessels, which correlates with higher expression and activity of the Notch pathway. Reduction of flow impaired endothelial Notch activity, bone angiogenesis and osteogenesis. In aged mice, type H capillaries, vessel buds, endothelial Notch signalling and osteogenesis were low, all of which were reactivated by induction of EC-specific Notch activity leading to significant increases in osteoprogenitors and mineralized bone.

laser Doppler perfusion imager (PIM II, Perimed, Sweden). Animals were euthanized at 16 or 48 hps and tibias were collected for analysis.

**Immunohistochemistry.** Freshly dissected tibiae were collected from wild type, mutant mice and their control littermates. Bone were processed and imaged as reported earlier[8,9]. The following primary antibodies were used: Emcn (sc-65495, Santa Cruz, diluted 1:100), Pecam1 conjugated to Alexa Fluor488 (FAB3628G, R&D Systems, 1:100), Pecam1 (553370, BD Pharmingen, 1:100), Osterix (sc-22536-R, Santa Cruz, 1:200), alphaSMA-Cy3 (C6198, Sigma, 1:100), Calcitonin receptor (ab11042, Abcam, 1:75), Dll4 (AF1389, R&D Systems, 1:100), Biotin-conjugated CD45 (553077, Becton Dickinson, 1:100) and Ter-119 (MAB1125, R&D Systems, 1:100). After sections had been washed with PBS for three times, the following secondary antibodies were used: Anti-rat Alexa Fluor 594 (Invitrogen; A21209, 1:400), anti-rabbit Alexa Fluor 546 (Invitrogen; A11035, 1:400), anti-goat Alexa Fluor 488 (Invitrogen; A11055, 1:400) and streptavidin Alexa Fluor 488 (Invitrogen; S11223, 1:400). Immunofluorescent stainings were analysed at high resolution with a Zeiss laser scanning confocal microscope, LSM780.

Proliferating cells were labelled by injecting mice intraperitoneally with EdU (Invitrogen) 3 h before euthanasia. Mice were killed; bones were collected and processed following the protocol mentioned before. For immunostaining EdU using Click-iT chemistry, manufacturer's protocols were followed.

**Image acquisition and analysis.** Z-stacks of images were processed and 3D reconstructed with the Imaris software (version 7.00, Bitplane). Imaris, Image J, Zen (Carl Zeiss), Photoshop and Illustrator (Adobe) softwares were used for image and movie processing in compliance with *Nature*'s guide for digital images. For all quantifications, a region of 300–400 μm from growth plate towards the caudal region was selected in the metaphysis. All quantifications were done with Image J software on high-resolution confocal images.

**Quantitative RT–PCR.** For transcript analysis, fresh bone samples were crushed and digested with Collagenase to prepare a single-cell suspension. Pure ECs were sorted by flow cytometry using a FACSAria (BD Biosciences) directly into the lysis buffer of RNeasy Mini Kit (Qiagen). Total RNA was isolated following the manufacturer's protocol and, subsequently, a total of 100–500 ng RNA was used to generate cDNA with the iScript cDNA Synthesis System (Bio-Rad). Further, quantitative PCR (qPCR) was performed using TaqMan gene expression assays on an ABI PRISM 7900HT Sequence Detection System. TaqMan Gene Expression Master Mix (Applied Biosystems) was used to perform qPCR for the FAM-conjugated TaqMan probes *Pecam1, Nos3, Dll4, Jag1, Hes1, Hey1, Hey2, Hes5, Angpt2, Cdh5, Sp7, Bglap, Tgfb1, Tgfb2, Tgfb3, Bmp2, Bmp4, Fgf1, Pdgfb* and *Wnt10b* commercially available at Life Technologies (Thermo Fisher Scientific). All the gene expression assays were normalized using endogenous VIC-conjugated *Actb* probes. For the analysis of *Bgalp* and *Sp7* expression, RNA was isolated from whole-bone samples. Freshly dissected bones were immediately flushed (to remove haematopoietic cells), crushed and lysed in lysis buffer for 15 min before proceeding for RNA isolation, cDNA preparation and qPCR analysis as described above.

**Bone blood flow measurements.** Blood flow measurements were performed in tibia of mice following a previously published protocol[75]. Briefly, anaesthetized mice were given intracardiac injection of fluorescent microspheres (Invitrogen FluoSpheres 15 μm, cat. no. F8842). Mice were killed after 1 min and bones (tibia, femur and sternum) were collected immediately. After decalcifying the bones for 24–48 h, they were subjected to ethanolic lysis for 1–2 days in a shaker incubator at 37 °C, 75 r.p.m. and protected from light. Then the samples were run through filtration unit to collect microspheres in a nylon membrane. Finally, spheres were collected in 1 ml of Cellosolve acetate and fluorescence intensity is quantified using microplate reader.

**Intravital imaging.** 10-day-old *Flk1-GFP* knock-in mice were anaesthetised using ketamine and xylazine. Following the careful dissection of skin and muscle covering the metatarsals, mice were placed on stage of a Zeiss LSM780 laser scanning confocal microscope. Depending on the region of interest, location of metaphysis or diaphysis was selected for further imaging. After 1 h, the animals were continuously provided with isoflurane and oxygen combinations. Blood vessel growth was monitored for 6–10 h continuously with every 5–10 min scanning depending on experimental requirements. Mice were euthanized with an overdose of anaesthetics after the experiment. All the images were analysed using ZEN 2010 and Imaris software platforms.

**Quantification of blood flow speed and shear rate.** Movies of red blood cell (RBC) movement in the bone vasculature were captured in the anaesthetized *Flk1-GFP* mice after dextran (2,000 kDa) injection using LSM780 confocal microscope with the scan speed of 120 frames per second. Line scan analysis was performed to trace the displacement of RBCs and calculate the speed of erythrocytes based on the distance RBCs displaced per time unit ( = number of frames × 1/120 s). Using this blood flow speed, shear rate was calculated considering blood vessels as straight cylinders and blood as an ideal Newtonian fluid with constant viscosity. Under these conditions, shear rate ($\gamma$) experienced by ECs lining a blood vessel is $8v/d$ where $v$ is the blood speed and $d$ is the diameter of the blood vessel.

**Flow cytometry.** Freshly isolated bone samples were cleaned thoroughly to remove the adherent muscles and then crushed in ice-cold PBS with a mortar and pestle. Whole bone marrow was digested with Collagenase at 37 °C for 20 min. After washing and filtering, single-cell suspensions were subjected to immunostaining. Cells were stained with Emcn (sc-65495, Santa Cruz, 1:100) antibody for 45 min. After washing in PBS, cells were stained with anti-rat APC (Jackson laboratories, 712-136-153, 1:50) and CD31-conjugated with Alexa Fluor 488 (R&Dsystems, FAB3628G, 1:50) for 45 min. After washing, data were acquired and analysed on BD FACSVantage cytometer (BD Biosciences).

**Micro-CT analysis and histomorphometry.** Tibiae were collected from mutants and their littermate controls, or from treated mice and sham controls. Bones were cleaned thoroughly to remove surrounding soft tissues and fixed in 4%

paraformaldehyde. The fixed tibiae were analysed using micro-CT ($\mu$CT 35) and software IPL V5.15 at Scanco Medical AG, Switzerland. A voxel size of 12 $\mu$m was chosen in all three spatial dimensions. For each sample, 148 from 232 slices were evaluated covering a total of 1.776 mm at a voltage of 70 kVp, intensity 114 $\mu$A and integration time 1,200 ms. Osteoclast number/bone perimeter (No. Oc./B. Pm) were calculated based on Calcitonin receptor staining of bone sections.

**Electron Microscopy (SEM and transmission electron microscopy).** Tibiae from 21-day-old mouse were dissected, cut open on their distal part and submerged in 4% paraformaldehyde, 0.5% glutaraldehyde, 2 mM MgCl$_2$, 2 mM CaCl$_2$ in 0.1 M cacodylate buffer, pH 7.4, under agitation for 2 h at room temperature. Samples were fixed further for 1 h in 2% paraformaldehyde, 2% glutaraldehyde, 2 mM MgCl$_2$, 2 mM CaCl$_2$ in 0.1 M cacodylate buffer, pH 7.4 and overnight at 4 °C. Next, bones were then decalcified over 5 days (with one change of solution after 2 days) in 5% EDTA in 0.1 M cacodylate buffer, pH 7.4 under rotation at 4 °C. Subsequently, 100 $\mu$m sections were generated with a vibratome (VT 1200, Leica, Viennna, Austria). Sections were post-fixed in 1% OsmO$_4$, containing 1.5% KFeCN in 0.1 M cacodylate buffer, pH 7.4. Sections for scanning electron microscopy (SEM) analysis were glued on coverslips with 4% low melting point-agarose and dehydrated stepwise in ethanol. Finally they were critical point dried (CPD 300, Leica, Vienna, Austria) with 35 exchanges against CO$_2$. Before examination, SEM samples were processed by sputtering with platinum and carbon. Analysis was performed at a Zeiss-Leo-SEM (NanoAnalytics, Münster, Germany). Vibratome-sections chosen for transmission electron microscopy were dehydrated and embedded flat in epon. Areas of interest were ultrathin (60 nm) sectioned and placed on a formvar-filmed single-copper grid. Samples were counterstained with uranyl-acetate and lead, and imaged at a transmission electron microscope at 80kV (Tecnai12-biotwin, FEI, Eindhoven, Netherlands).

**Experiments using cultured ECs.** Human umbilical vein ECs were obtained from Lonza (cat. No. CC-2529). Primary ECs were cultured in M199 medium supplemented with 1:600 bovine brain extract (Lonza, Braine-l'Alleud, Belgium), a cocktail of penicillin, streptomycin and Amphotericin B (Invitrogen, Life Technologies, Ghent, Belgium), 10 kU heparin (Sigma-Aldrich, Bornem, Belgium) and 20% FBS (Merk Millipore, Berlin, Germany). Cells were seeded and propagated on 0.1% gelatin (Sigma-Aldrich, St Louis, MO) coated dishes and incubated at 37 °C in a 5% CO$_2$ humidified incubator.

For shear stress experiments, human umbilical vein ECs were seeded on human fibronectin (Corning Lasne, Belgium) coated slides at a density of 500,000 cells/slide and let to grow for 3 days. The day of the experiment, the slide containing the cell monolayers was inserted in a parallel plate flow chamber designed in-house, the chamber was connected to a perfusion system containing a media reservoir and a peristaltic pump (Cole Parmer, Antwerpen, Belgium). ECs were exposed to laminar flow at a defined shear stress of 15 dyn cm$^{-2}$ for 4 h for protein extraction or 24 h for RNA extraction. For each experiments cells were cultured in static condition. As negative control for NICD generation, cells were treated with 50 $\mu$M DAPT or DMSO (vehicle) from 24 h prior exposure to shear stress until the end of the experiment.

For Western blotting, RIPA buffer (Thermo Scientific, Bremen, Germany) was used to extract proteins, supplemented by a cocktail of protease and phosphatase inhibitors (Roche, Vilvoorde, Belgium). Protein separation was performed under reducing conditions on NuPAGE gel 3–8% Tris-Acetate (Thermo Scientific, Bremen, Germany). Proteins were transferred to a to a nitrocellulose membrane. 5% milk was used as blocking buffer for 1 h followed by incubation with primary antibody in TBST 5% BSA overnight at 4 °C or room temperature. After washing in TBST, the membrane was incubated with secondary antibody for 1 h at room temperature and signal was detected with Pierce ECL Western Blotting Substrate (Thermofisher Scientific, Waltham, MA). Images were cropped and assembled with ImageJ software. NICD was detected with rabbit anti-cleaved Notch1 antibody (#4147, Cell Signaling Technology, Danvers, MA, 1:1,000). Rabbit anti-β-actin (#4970, Cell Signaling Technology, 1:2,000) was used as loading control.

For reverse transcription quantitative PCR (RT-qPCR), isolation of the total RNA was performed using TRIzol reagent (Invitrogen Life Technologies, Carlsbad, CA), and cDNA was synthetized with High-Capacity cDNA Reverse Transcription Kit (Thermofisher Scientific, Waltham, MA). RT-qPCR was performed with an ABI 7500HT fast real-time PCR System (Applied Biosystems, Foster City, CA) using SYBR Select Master Mix (Thermofisher Scientific, Waltham, MA). Selected genes: HES1 (NM_005524.3), HEY1 (NM_012258.3), HEY2 (NM_012259.2), JAG1 (NM_000214.2) and NOTCH1 (NM_017617.3). HPRT1 (NM_000194.2) was used for normalization.

To analyse the effect of Alendronate on bone endothelium, primary bone ECs from wild-type mice were isolated and cultured following a protocol described earlier[8]. Cultures were maintained and passaged minimum twice before the experiment. Cells were treated with Alendronate (1 $\mu$M) for 8 h and subsequently lysed to isolate RNA and perform qPCR as described before.

**Statistics.** All biological parameters are presented as mean ± s.d. and counting data as mean ± s.e.m. The significance of difference in the mean values was determined using two-tailed Student's $t$ test unless otherwise mentioned. $P < 0.05$ was considered significant. All calculations were performed using GraphPad Prism

software. No randomization or blinding was used, and no animals were excluded from analysis. Sample sizes were selected on the basis of previous experiments. Several independent experiments were performed to guarantee reproducibility of findings.

**Data availability.** All relevant data are presented in the manuscript and the supplementary file and available from the authors upon request.

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

## Acknowledgements

We thank Stefan Volkery for helping with imaging, Martin Stehling for flow cytometry and Sylvana Henschen for femoral artery ligations. Funding was provided by the Max Planck Society, the University of Münster and the DFG cluster of excellence 'Cells in Motion' (S.K.R. and R.H.A). The European Research Council supported this project with the Advanced Grant 339409 (AngioBone; R.H.A) and the Consolidator Grant 64757 (rEnDOx; M.M.S.).

## Author contributions

S.K.R., A.P.K. and R.H.A. designed the experiments and interpreted results. S.K.R. and A.P.K. developed intra vital-imaging of long bones. D.Z. performed electron microscopy, C.M. and M.M.S. the in vitro flow experiments. J.G. and F.P.L. generated femoral artery ligations. S.K.R. and A.P.K. directed M.S., performed all other experiments, generated and characterized mouse mutant lines. M.G.B provided reagents. A.M. provided Flk1-GFP mice. S.K.R., A.P.K. and R.H.A. wrote the manuscript.

## Additional information

Competing financial interests: The authors declare no competing financial interests.

