## [Peer Review File · Nature Communications]

Reviewers' comments:

Reviewer #1 (expert in angiogenesis and Notch signaling)

Remarks to the Author:

Ramasamy et al., show very interesting novel data about blood flow dynamics in long bone and how this is linked to Notch signaling in endothelial cells and angiogenesis. The work shows fundamental new aspects of angiogenesis by endothelial buds close to the growth plate in bone. This appears to be strongly dependent on proper blood flow as shown by femoral artery ligation and treatment with Prazosin and Angiotensin-2.

Major points:

1) The manuscript is too diverse. One should focus on this novel type of angiogenesis in long and the functional link to blood flow.

2) Any mechanistic data how blood flow in bones affects Notch signaling is missing. This is in my view the weakest aspect of this work. Different types of shear stress change the expression of Notch target genes in cultured ECs. Careful evaluation of the literature, combined with detailed analysis which hemodynamics act on bone ECs at the EC buds might help to figure out this issue.

3) The data showing that Prazosin impairs bone angiogenesis and mineralization are very convincing. The data regarding Clonidine treatment are not. Quantification of EC buds should be performed. Furthermore, the authors do not explain why they use an alpha1 receptor inhibitor to decrease blood flow in bone. Is this a specific effect or secondary to lowering blood pressure.

4) The data about increased angiogenesis after treatment with bisphosphonates is exciting. It is however not clear if altered bone mineralization, stiffness etc. after bisphosphonate application affects blood vessels or if there is also a novel action directly on endothelial cells. The authors show some data about increased blood flow a few hours after drug administration. However, this is not really a proof for their hypothesis and any additional mechanistic insights are completely lacking. From a clinical point of view it is not clear why the authors use young mice for treatment. These drugs are usually given to postmenopausal women.

5) It is fantastic to see that endothelial Notch1-ICD expression is sufficient to reactivate angiogenesis and osteogenesis in old bone. However, there is no evidence how this could be achieved as a potential treatment. Again, this makes the manuscript even more diverse.

Reviewer #2 (expert in mechanotransduction in vessels)

Remarks to the Author:

The authors have shown that the NOTCH pathway has a key role in the control of angiogenesis and osteogenesis by flow (shear stress).

This study is done properly and addresses a major question; i.e. mechanosensitive pathway in angiogenesis and osteogenesis;

Nevertheless, several concerns remain:

1- Blood flow velocity in the various vessel types: shear should also be provided, as it is the force acting on the endothelial cells.

2- Blood flow alteration in bones: ligating the femoral artery also induces inflammation, oxidative stress, hypoxia in the surrounding tissues (mainly the skeletal muscle) and this is likely to influence

bone homeostasis in addition of blood flow per se. any way to alter bone blood flow without altering the surrounding tissues?

3- Prazosin, a known alpha-adrenergic and shear-stress inhibitor: I do not agree with this as prazosin is expected to increase arterial diameter and most probably increase blood flow and consequently shear stress. Is there articles supporting this assumption; may be in bones? To the best of my knowledge, prazosin is accelerating flow and consequently arteriogenesis and angiogenesis. Did the author measure the effect of prazosin (and clonidine) on blood flow in bone vessels (acute and chronic effects?). Alternatively, pressure reduction might cause the effect observed. The use of a drug not affecting the adrenergic system would help solve the problem.

4- 12-week old mice may not be totally mature. For example, blood pressure stabilizes latter than this in rats and mice leading to errors when talking of aging of the vasculature comparing 12-week old mice to older (1-2 year old) mice. 5-6-month old mice might be better "adult" control mice.

Reviewer #3 (expert in osteogenesis and bone physiology)

Remarks to the Author:

This is an interesting manuscript following a series of studies by the authors on the coupling of angiogenesis and osteogenesis. The authors first demonstrate vascular organization and pattern in long bone by immunostaining and high-resolution confocal microscopy. They then investigated the how blood flow regulates bone angiogenesis through femoral artery ligation and the use of prazosin (a known alpha-adrenergic inhibitor) and clonidine (another flow-reducing drug). The experiments show that the inhibition of local blood flow strongly reduces numbers of columnar vessels and endothelial bud structures in the proximity of the growth plate, without promoting endothelial apoptosis. They also find that the arches and buds are dynamic during active angiogenesis and mediate vessel growth in long bone. The authors also demonstrate that local blood flow mediates the coupling of angiogenesis and osteogenesis via endothelial Notch signaling in bone. Interestingly, their work also shows that the reactivation of type H vessels by the Notch pathway in aged mice can promote both angiogenesis and osteogenesis leading to improved bone mass. Finally, treatment with bisphosphonate, a widely use anti-resorptive drug used for osteoporosis, increases blood flow and angiogenesis in bone.

Comments: This is a well done study which is likely to have a significant impact not only in our understanding of bone biology, but also interfaces with vascular biology. The experiments are nicely designed with appropriate controls, and the interpretation of the results seem very reasonable. Several issues need consideration.

1. There is no discussion on the role of vascular autocooids, such as nitric oxide, in vasoregulation in bone. These studies have been published way back in the 1990s (MacIntyre, ONAS, 1991 and others) and should be discussed in relation to these findings, particularly as the authors demonstrate changes in bone mass. Likewise the effect of peptides, such as CGRP, that have been shown to be osteoregulatory need to be discussed.

2. If possible, please address a potential mechanism of the effect blood flow on coupling of angiogenesis and osteogeneis, in addition to a nutrition supply. It would be interesting to focus on bone formation by Notch mediated angiogenesis (see recent paper by Liu et al, PNAS).

3. Are there any data available that shows effect of bisphosphonate on isolated bone endothelial cells in cell culture?

4. Correction of a typo 'loalized' on Page 6, Line5.

We thank all the referees for their time and useful feedback. All comments were very useful and constructive. As you will see, the suggestions have been incorporated into the revised manuscript, which has undoubtedly improved the manuscript.

Reviewer #1 (expert in angiogenesis and Notch signaling)

Remarks to the Author:

Ramasamy et al., show very interesting novel data about blood flow dynamics in long bone and how this is linked to Notch signaling in endothelial cells and angiogenesis. The work shows fundamental new aspects of angiogenesis by endothelial buds close to the growth plate in bone. This appears to be strongly dependent on proper blood flow as shown by femoral artery ligation and treatment with Prazosin and Angiotensin-2.

We would like to thank the reviewer for the positive feedback.

Major points:

1) The manuscript is too diverse. One should focus on this novel type of angiogenesis in long and the functional link to blood flow.

We agree that the study is addressing different aspects of the crosstalk between flow, growth of the bone vasculature, Notch and osteogenesis. However, all these points are also closely interrelated and this is now explained in a more logical and more readily understandable fashion in the revised manuscript. As suggested by the reviewer, strong emphasis has been placed on the novel type of angiogenesis, the functional link to blood flow and the role of Notch signaling.

2) Any mechanistic data how blood flow in bones affects Notch signaling is missing. This is in my view the weakest aspect of this work. Different types of shear stress change the expression of Notch target genes in cultured ECs. Careful evaluation of the literature, combined with detailed analysis which hemodynamics act on bone ECs at the EC buds might help to figure out this issue.

As suggested by the reviewer, we have addressed this critical question. In particular, we have added new data showing that flow-controlled genes (namely Klf2, Nos3 and Pecam1) are downregulated in Notch loss-of-function mutants (new data in Supplementary Fig. 8a), which reflects the previously reported dilation and disorganization of bone vessels in these mutants (Ramasamy et al., Nature 2014). Conversely, Klf2, Nos3 and Pecam1 transcript levels were increased in bone ECs isolated Notch gain-of-function mutants (Supplementary Fig. 8a).

Other new data show that flow enhances Notch activation and Notch target gene expression in cultured endothelial cells (Supplementary Fig. 8d, e). This

particular experiment was conducted with HUVECs because a similar experiment with bone ECs is currently technically not possible. Based on the sum of the data, we propose that Notch signaling is, at least in part, controlled by flow. At the same time, Notch controls flow and shear stress in the bone endothelium by controlling angiogenesis and the morphology of the local vasculature. We have also extended the discussion of the relevant literature on the interplay between flow and the Notch pathway (see page 14-16 of the revised manuscript).

3) The data showing that Prazosin impairs bone angiogenesis and mineralization are very convincing. The data regarding Clonidine treatment are not. Quantification of EC buds should be performed. Furthermore, the authors do not explain why they use an alpha1 receptor inhibitor to decrease blood flow in bone. Is this a specific effect or secondary to lowering blood pressure.

Agree. As suggested by the reviewer, we have performed the quantitation of EC buds after Clonidine treatment (Supplementary Fig. 3h), which shows a significant reduction in the number of endothelial buds. We also provide a better justification for the use of Prazosin on page 6 (middle paragraph) of the revised manuscript.

4) The data about increased angiogenesis after treatment with bisphosphonates is exciting. It is however not clear if altered bone mineralization, stiffness etc. after bisphosphonate application affects blood vessels or if there is also a novel action directly on endothelial cells. The authors show some data about increased blood flow a few hours after drug administration. However, this is not really a proof for their hypothesis and any additional mechanistic insights are completely lacking. From a clinical point of view it is not clear why the authors use young mice for treatment. These drugs are usually given to postmenopausal women.

We thank the reviewer for raising this important point. The role of bisphosphonates in bone mineralization is well established, but the relation to blood vessels and angiogenesis was unknown. While our proof-of-principle experiment shows that Alendronate triggers a robust vascular response and enhances flow in vivo, newly added data shows that the drug does not affect activity of the Notch pathway in cultured bone endothelial cells (Supplementary Fig. 10f). Thus, we are looking at systemic effects that might involve flow changes or molecular interactions between multiple cell types. We would also like to clarify that the experiments shown in Fig. 6b-d and 6f, g were performed with aged animals, which is also indicated by the labels in Fig. 6b.

5) It is fantastic to see that endothelial Notch1-ICD expression is

sufficient to reactivate angiogenesis and osteogenesis in old bone. However, there is no evidence how this could be achieved as a potential treatment. Again, this makes the manuscript even more diverse.

We agree that this is a very important and exciting finding. If similar mechanisms are at play in the human skeletal system, the vasculature could be promising therapeutic target for the treatment of bone loss. It is also unquestionable that further studies, such as a validation of findings in human bone or the development of treatment leading to (ideally EC-specific) Notch activation, are crucial for the exploration of such therapeutic opportunities. Nevertheless, we see the current study as an important milestone because it provides the proof-of-principle data that EC-specific Notch activation can induce osteogenesis in the aged organism.

Reviewer #2 (expert in mechanotransduction in vessels)

Remarks to the Author:

The authors have shown that the NOTCH pathway has a key role in the control of angiogenesis and osteogenesis by flow (shear stress). This study is done properly and addresses a major question; i.e. mechanosensitive pathway in angio and osteogenesis;

We are grateful to the reviewer for the constructive comments and suggestions, which helped us to improve the manuscript.

Nevertheless, several concerns remain:

1- Blood flow velocity in the various vessel types: shear should also be provided, as it is the force acting on the endothelial cells.

We agree with the comment. Calculation of the shear rate acting on the type H and type L endothelial cells in bone is now shown in Supplementary Fig. 2c. The calculated shear rate for type H capillaries is relatively low (900s^{-1}), which corresponds to a wall shear stress of 31.5 dynes/cm^2 .

2- Blood flow alteration in bones: ligating the femoral artery also induces inflammation, oxidative stress, hypoxia in the surrounding tissues (mainly the skeletal muscle) and this is likely to influence bone homeostasis in addition of blood flow per se. any way to alter bone blood flow without altering the surrounding tissues?

We agree that this is an intrinsic and inevitable issue of this type of experiment. To minimize the impact of such secondary effects, we analyzed bones at 16 hours after ligation. Interestingly, apoptosis (active Caspase3) was not induced at this time point, whereas the increased anastomosis of buds was already visible (Suppl. Fig. 3b-d).

Pharmacological drugs were used as an alternative approach for the modulation of blood flow in bone. While it is appreciated that these experiments have their own intrinsic limitations, both strategies – ligation and pharmacological flow modulation – yielded very consistent results.

3- Prazosin, a known alpha-adrenergic and shear-stress inhibitor: I do not agree with this as prazosin is expected to increase arterial diameter and most probably increase blood flow and consequently shear stress. Is there articles supporting this assumption; may be in bones? To the best of my knowledge, prazosin is accelerating flow and consequently arteriogenesis and angiogenesis. Did the author measure the effect of prazosin (and clonidine) on blood flow in bone vessels (acute and chronic effects?). Alternatively, pressure reduction might cause the effect observed. The use of a drug not affecting the adrenergic system would help solve the problem.

4- 12-week old mice may not be totally mature. For example, blood pressure stabilizes latter than this in rats and mice leading to errors when talking of aging of the vasculature comparing 12-week old mice to older (1-2 year old) mice. 5-6-month old mice might be better "adult" control mice.

It is correct that one would have expected an increase in local flow after Prazosin administration due to an increase in arterial diameter. However, our actual measurements (Fig. 2c) show decreased bone blood flow and downregulated expression of flow-controlled genes in bone ECs of Prazosin-treated animals (Suppl. Fig. 3f). This effect might be caused by reduced blood pressure and/or the diversion of flow into other organ systems.

To our knowledge, there are no previous studies that have analyzed the effect of Prazosin on flow in the skeletal system. There is, however, one recent paper showing that systemic administration of Prazosin leads to impaired bone formation (Tanaka et al. 2016, Br. J. Pharmacol. 173:1058-69). This result is consistent with our own observations regarding the effect of Prazosin on bone angiogenesis, Notch signaling status (Fig. 5a, b), the expression of potential angiocrine signals (Fig. 4f), and osteogenesis (Fig. 4c-e; Suppl. Fig. 6a-d). As in many of our studies on bone vessels, we prefer to use younger animals because the fraction of type H vessels and ECs is already strongly reduced in adult mice. Likewise, it is much more straightforward to study angiogenic vessel growth, bud formation and anastomosis in younger animals.

Reviewer #3 (expert in osteogenesis and bone physiology)

Remarks to the Author:

This is an interesting manuscript following a series of studies by the authors on the coupling of angiogenesis and osteogenesis. The authors first demonstrate vascular organization and pattern in long bone by immunostaining and high-resolution confocal microscopy. They then investigated the how blood flow regulates bone angiogenesis through

femoral artery ligation and the use of prazosin (a known alpha-adrenergic inhibitor) and clonidine (another flow-reducing drug). The experiments show that the inhibition of local blood flow strongly reduces numbers of columnar vessels and endothelial bud structures in the proximity of the growth plate, without promoting endothelial apoptosis. They also find that the arches and buds are dynamic during active angiogenesis and mediate vessel growth in long bone. The authors also demonstrate that local blood flow mediates the coupling of angiogenesis and osteogenesis via endothelial Notch signaling in bone. Interestingly, their work also shows that the reactivation of type H vessels by the Notch pathway in aged mice can promote both angiogenesis and osteogenesis leading to improved bone mass. Finally, treatment with bisphosphonate, a widely used anti-resorptive drug used for osteoporosis, increases blood flow and angiogenesis in bone.

Comments: This is a well done study which is likely to have a significant impact not only in our understanding of bone biology, but also interfaces with vascular biology. The experiments are nicely designed with appropriate controls, and the interpretation of the results seems very reasonable. Several issues need consideration.

We are very grateful for the positive feedback and thank the reviewer for the comprehensive analysis and valuable suggestions, which have helped us to improve the manuscript.

1. There is no discussion on the role of vascular autocooids, such as nitric oxide, in vasoregulation in bone. These studies have been published way back in the 1990s (MacIntyre, PNAS, 1991 and others) and should be discussed in relation to these findings, particularly as the authors demonstrate changes in bone mass. Likewise the effect of peptides, such as CGRP, that have been shown to be osteoregulatory need to be discussed.

As suggested by the reviewer, we now mention these studies in our Discussion (see page 14 of the revised manuscript).

2. If possible, please address a potential mechanism of the effect blood flow on coupling of angiogenesis and osteogenesis, in addition to a nutrition supply. It would be interesting to focus on bone formation by Notch mediated angiogenesis (see recent paper by Liu et al, PNAS).

It is correct that Notch signaling plays important roles in the bone endothelium but also in osteoblast lineage cells. The surprising and exciting findings made by Liu et al. (PNAS USA 2016) are now discussed on page 14/15 of the revised manuscript.

3. Are there any data available that shows effect of bisphosphonate on isolated bone endothelial cells in cell culture?

To our knowledge, there is no published data showing direct effects of bisphosphonate on isolated (cultured) bone endothelial cells. We have therefore performed this experiment and found that bisphosphonate treatment did not alter the expression of Notch pathway genes in cultured bone endothelial cells (Supplementary Fig. 10f).

4. Correction of a typo 'loalized' on Page 6, Line5.

Thank you. This mistake has been corrected.

REVIEWERS' COMMENTS:

Reviewer #1 (Remarks to the Author):

none

Reviewer #2 (Remarks to the Author):

The authors have responded satisfactorily to my comments.

The study is of major importance in the domain of osteogenesis and angiogenesis.

Reviewer #3 (Remarks to the Author):

The authors have taken into account all criticisms and have addressed all concerns.